# Beaver Dams and Fallen Trees as Ecological Corridors Allowing Movements of Mammals across Water Barriers—A Case Study with the Application of Novel Substrate for Tracking Tunnels

**DOI:** 10.3390/ani13081302

**Published:** 2023-04-11

**Authors:** Zuzanna Wikar, Mateusz Ciechanowski

**Affiliations:** Department of Vertebrate Ecology and Zoology, University of Gdansk, Wita Stwosza 59, 80-308 Gdansk, Poland; mateusz.ciechanowski@ug.edu.pl

**Keywords:** *Castor fiber*, ecosystem engineer, keystone species, habitat connectivity, new method

## Abstract

**Simple Summary:**

To cross barriers in their habitats, animals often use specific corridors, and some of them may be created by other species, such as beavers (*Castor canadensis* and *Castor fiber*). Their dams on rivers may act as bridges for land mammals, but their importance is largely unknown. We investigated the function of beaver dams using tracking tunnels with kinetic sand to collect animal tracks. We assessed its suitability for this purpose since it has never been used as a tracking medium before. We placed those tunnels on dams, fallen trees (logs), and floating rafts and found that kinetic sand perfectly preserved the tracks of small carnivores, allowing easy identification, whereas shrews and rodents smaller than rats could only be detected but not recognised. The highest activity of mammals was observed on dams, as they may provide shelter, which allows safe travel and even residence. A slightly higher diversity was found on logs due to the visits of carnivores, which prefer leaving their scats in exposed places as markings. Our results reveal another function of the beaver as a creator of habitats for other animals and provide a novel tool for monitoring mammal activity.

**Abstract:**

Physical obstacles within animal habitats create barriers to individual movements. To cross those barriers, specific corridors are used, some of them created by keystone species such as Eurasian beavers (*Castor fiber*). Their dams on rivers may also increase habitat connectivity for terrestrial mammals, but the significance of that function has never been quantified. To investigate this, we placed tracking tunnels on beaver dams, fallen trees, and—as a control—on floating rafts. Additionally, we tested kinetic sand as a novel substrate for collecting tracks and found the paws of small mustelids precisely imprinted in that medium, allowing easy identification. However, we needed to lump all shrews and rodents smaller than water voles (*Arvicola amphibius*) into one category as they can only be detected but not identified. The highest mammalian activity was observed on dams, as they may provide shelter, offering protection from predators during a river crossing or permanent residence, and even the opportunity to hunt invertebrates. Slightly higher diversity was found on logs because of a higher proportion of mustelids, which select exposed locations for scent marking. Our results increase our body of knowledge about the beaver as an ecosystem engineer and provide a novel tool for the monitoring of mammal activity.

## 1. Introduction

Home ranges of highly mobile terrestrial animals are rarely used evenly by their respective individuals, which usually aggregate their presence and activity in specific habitat patches that provide crucial resources [1]. The level of that space use heterogeneity, however, relates directly to habitat heterogeneity, which affects home range size and reveals significant interspecific variation [2]. Movements among those preferred patches are often optimised to reduce energy expenditure. However, in landscapes with more complex topographies, it does not always mean that they follow the shortest distances [3]. As in such landscapes, many physical obstacles to animal movements may occur; corridors that allow crossing these barriers often act as another group of structures where the relative level of an individual’s activity is disproportionately high compared to the total share of those structures in the whole area of the home range [4,5]. Some of these corridors might develop as a result of geomorphological or hydrological processes, e.g., fords on rivers [6], narrow sections of rivers with low discharge [7], or mountain passes [8], whereas others are created by moving individuals themselves, such as the paths of large ungulates in dense forest undergrowth [9]. The latter case may also exemplify a situation in which a moving animal acts as a keystone species/ecosystem engineer that increases landscape connectivity by either removing physical obstacles or building structures that allow the other animals to cross the barrier. Ecosystem engineers are ubiquitous in terrestrial and wetland habitats around the world [10], yet documentation of their function as providers of movement corridors for other species is surprisingly scarce [11].

The Eurasian beaver *Castor fiber* is a keystone species that has a disproportionate effect on the functioning and structure of ecosystems. It is one of the few species, apart from humans, capable of significantly transforming them, both in terms of geomorphology and hydrology [12]. The impact of the beaver, primarily in the form of slowing down the flow of water and creating ponds that enrich habitats, covers many groups of organisms—plants [13], lissamphibians [14], birds [15], but also other mammals [16,17]. The specific feature of beaver activity is the building of relatively large structures, i.e., lodges and dams, that may act as both nexuses and corridors for the movements of other mammals. That applies especially to semiaquatic species, such as the muskrat *Ondatra zibethicus* or the American mink *Neogale vison* [18], which can use lodges as permanent shelters [19,20]. Research conducted in Belarus has shown that the European mink *Mustela lutreola* is strongly associated with beavers and uses both lodges and beaver ponds. The occurrence of the American mink is also related to that of beavers, and in addition to the use of lodges as shelters, minks also hunt in lodges and tunnels dug by beavers [21]. The activity and species richness of small mammals from the orders of rodents (Rodentia) and shrews (Soricidae: Soricomorpha) is higher in lodges than in habitats without beavers, which proves that these are important facilities for these animals [22].

Contrary to lodges, there is a lack of published research on the importance of beaver dams for other animals, especially data on the species composition and frequency of the use of these structures by other mammals. Earlier work has focused on the role of beaver ponds and beaver-transformed habitats, not on the importance of dams as a structure. During live trapping, small mammals were often caught on the dams themselves and in the immediate vicinity near them [23]. The only work describing the use of beaver dams as bridges by mammals is based on an extremely small sample (two dams), and the lack of control points excluded testing the hypotheses about the selection of dams as places to cross watercourses [24]. As beaver dams usually cross water courses perpendicularly, one might expect that they act as corridors, allowing terrestrial animals to cross aquatic barriers without the necessity to swim; however, this function is yet to be quantified. Small forest rivers are also abundant in tree logs connecting two banks, both as a result of beaver foraging and other natural factors (e.g., windbreaks), providing an opportunity to test if beaver dams really improve habitat connectivity for other species. No study, however, has quantified the use of both logs connecting stream banks and beaver dams by other mammals at the same sites.

Tracking is one of the widespread methods applied to monitor space use by terrestrial mammals. If the conditions of a site or season do not provide any natural substrate that allows imprinting any tracks (snow, mud, sand), special tracking tunnels, pads, plates, or rafts are installed in the field, usually containing sooted metal sheets [25], acetate sheets with a graphite, alcohol, and oil coating [26], ink and paper [27], or wet clay [28]. Such devices are frequently used to monitor the use of wildlife crossings under motorways [29] and might thus also be applied to quantify animal movements across natural barriers. Despite the development of trail cameras in recent years, followed by a significant increase in the number of studies based on that technology [30], tracking pads still provide a cheap and environmentally friendly alternative for some research goals when detailed information about individual behaviour is not necessary. Trail cameras, which are several orders of magnitude more expensive, may cause significant financial losses in the budget of any research project if they are stolen from the field on a larger scale. Additionally, they become burdensome e-waste when they stop functioning or get damaged. In contrast, tracking pads and tunnels do not suffer from these drawbacks, making them a more sustainable solution Due to its specific physical properties, kinetic sand seems to be a promising medium for use in tracking tunnels. However, no study has ever applied it, nor is it considered in available guidelines for the monitoring of mammals.

This research aims to determine the role of beaver dams and tree logs as corridors that allow other species of mammals to cross watercourses, with an emphasis on small carnivores, rodents, and shrews. We hypothesised that the use of beaver dams and the species richness of mammals using them are higher than in the case of fallen tree logs. Additionally, we tested the suitability of kinetic sand as a substrate in tracking tunnels.

## 2. Materials and Methods

### 2.1. Study Area

We conducted the study in the Pomeranian Lakeland and the Baltic Sea Coast, Northern Poland. The area comprises mostly young, postglacial landscapes, and the land cover is dominated by a mosaic of arable land and meadows (~50%), forests, mostly managed and composed predominantly of Scotch pine (*Pinus sylvestris*), beech (*Fagus sylvatica*), and pedunculate oak (*Quercus robur*) (~36%), and human settlements (~6%). A dense network of lakes and small rivers covers ~5% of the region. Three small rivers were chosen as study sites (Figure 1):Swelinia River near the city of Sopot. Its entire drainage system is located in the urban or suburban setting (Tricity agglomeration), but only short sections, adjacent to densely built-up districts, are subject to hydromodification. It is a rapid stream of sub-montane characteristics with numerous meanders and cascades. Its banks are covered mostly by alder-ash riparian forest *Fraxino-Alnetum*, dominated by black alder *Alnus glutinosa*; the stand composition is enriched by several species growing in abandoned allotments, especially fruit trees.Source section of the Trzebiocha River near the city of Kościerzyna, between lakes Księże and Osuszyno. In contrast to the previous site, this one is located in a rural woodland setting. The riverbed is straight, and the current is slow. The banks are covered by managed forest, dominated by Scotch pine (*Pinus sylvestris*) with an admixture of spruce (*Picea abies*) and birch (*Betula pendula*), but the stands are subjected to more flooding due to beaver dams, which represent alder carr (*Sphagno squarrosi–Alnetum*).Górzynka River near Podole Wielkie village, also in a rural and woodland setting. The river is also rapid and has sub-montane characteristics, but its bed is much broader than that of the Swelinia River. Steep valley slopes almost reach the river in many spots; they are covered either by oak-hornbeam forest *Stellario-Carpinetum*, dominated by hornbeam *Carpinus betulus* and beech *Fagus sylvatica,* or young, dense spruce plantations. In places with extensive flooding because of beavers or narrow riverine terraces, patches of the alder-ash riparian forest also occur.

### 2.2. Experimental Design

To investigate the importance of beaver dams as communication routes for smaller semiaquatic and terrestrial mammals compared to the importance of tree logs, we used tracking tunnels with kinetic sand as a medium to collect the tracks of animals crossing the watercourse. Each tray was 45 × 35 cm, and we used 1 kg of kinetic sand per tunnel, resulting in a layer with an approximate thickness of 1 cm. To protect the kinetic sand from rain, we used PVC sheets, formed into arches, and fixed them to the trays. The peak of an arch was approximately 20 cm high, and it limited the size of the animals crossing the tray, which protected the tracks of the smaller mammals from being trampled by larger animals (Figure 2). Control points were floating rafts made from wooden pallets with tracking tunnels fixed in the centre. Tracking tunnels at the dams and tree logs were placed only in cases of full connectivity between riverbanks, and we therefore only selected the active dams without breaches or water flowing over the edge and trees that were over the water along its entire width. We used 8 tracking tunnels on dams, 11 on trees, and 12 on rafts (Table 1).

### 2.3. Analysis

Animal tracks left in kinetic sand were identified to the lowest possible taxon using identification keys [31,32,33]. All tracks were counted because it was impossible to separate individual tracks, and due to the occurrence of single tracks, activity was measured by the number of all paw prints. Due to the significant loss of tracking tunnels and consequent reduction in sample size, we merged the data from all three sites. By multiplying the length of the exposure and the number of tunnels in the given categories (dams, logs, and rafts), we obtained the number of trap days, which was used to calculate the average number of tracks.

We compared the species diversity on the dams and trees by calculating the Shannon-Wiener coefficient and tested the difference between the coefficients using Hutcheson’s t-test. The second method of assessing the diversity of mammals visiting tracking tunnels was to plot rarity curves for the number of individuals and the number of controls in the PAST 4.07b programme [34].

We compared the use of dams, logs, and rafts using the chi-square frequency test. The null hypothesis assumed no differences between the obtained and expected number of tracks in each category. We calculated the latter by multiplying the total number of tracks of all taxa in all categories by the proportion of the number of trap days on dams, trees, and controls (rafts). This procedure was necessary due to the different numbers of tunnels in individual categories and the different lengths of the exhibition. First, we tested the differences in the numbers of tracks obtained between the categories of tracks without any adjustments. However, all examined trees turned out to be narrower than the tunnels, and all dams were wider. It can therefore be assumed that only tunnels mounted on logs captured all small mammals crossing them, whereas the dams left space for animals to bypass the tracking tunnel. Thus, we decided to also test the difference between the use of dams and tree logs by mammals after adjusting for the number of tracks. For this purpose, we established a formula to estimate the number of tracks on dams:$$adjusted\;number\;of\;tracks=\frac{dam\;width}{track\;tunnel\;width}\times raw\;number\;of\;tracks$$

We re-adjusted the dam scores with the raw log and raft scores, computed new expected track numbers for all categories, and tested the null hypothesis a second time.

## 3. Results

Among the 31 tracking tunnels installed in the study sites, we obtained data from 18 of them, including 10 on tree logs, 4 on dams, and 4 on rafts; the remaining ones were lost. Two tunnels on the dams had been removed by humans, and two had been completely blocked by beavers by covering them with new construction material. Tracking tunnels on rafts had been sunk, probably by beavers, but at least two had been destroyed by humans. The only tracking tunnels at the tree log, from which data could not be retrieved, were flooded. The total exposure of individual categories of the tunnel was as follows: 1017 trap days on trees, 404 trap days on dams, and 180 trap days on rafts.

We identified the activity of at least five species of mammals. Among them, at least three of them represented the order Carnivora (European polecat *Mustela putorius* or American mink—Figure 3; the stoat *Mustela erminea*—Figure 3 and Figure 4); one larger rodent (water vole *Arvicola amphibius*—Figure 3 and Figure 5; in one case, similar tracks could also belong to the brown rat *Rattus norvegicus*); however, most of them were small rodents, or Soricomorphs, which could not be identified and were therefore classified into the collective category of small mammals, micromammalia (Figure 4).

In total, we assigned the tracks to six categories, of which six (at least five species) were found on logs, five (also at least five species) on dams, and only one (only micromammalia) at control points designated by floating rafts (Appendix A). We observed a significant difference in species composition between dams and trees (chi square = 21.91; *p* = 0.0002). Taking into account the proportions among individual taxa, logs were more willingly used by larger mustelids—polecats/minks and stoats—and dams by rodents and shrews. Most tracks were recorded on logs, more than half less on dams, and the least on rafts (Figure 6). The highest average number of tracks was found on dams (0.73 tracks/trap day), followed by logs (0.62 tracks/trap day), and rafts (0.12 tracks/trap day).

The value of the Shannon–Wiener diversity index for trees was 1.57, whereas, for dams, it was 1.27. The difference in diversity was highly significant (t = 3.34, df = 501.72, *p* = 0.0009). Rarefaction curve analysis showed that the species diversity of mammals was slightly higher in trees than in dams, indicating that with the currently available accuracy of species determination, the list of mammal taxa using logs and dams was almost complete—they reached a plateau already with 75–100 tracks with the value of S ≈ 5 species found by us (Figure 7). Therefore, one should not expect a particular increase in species richness along with a further increase in the sample.

There were significant differences in the activities of mammals among all categories of tracking tunnels (chi square = 61.66; *p* < 0.0001). In the case of results not adjusted for the width of dams, the actual number of tracks recorded on trees and dams was higher than expected, whereas it was lower on rafts (Figure 8). However, despite the greater number of tracks in trees, we found no significant difference between activity in trees and dams (chi square = 2.53; *p* = 0.11). We showed significant differences by comparing dams and rafts (chi square = 59.94; *p* < 0.0001) and trees and rafts (chi square = 53.54; *p* < 0.0001).

When we accounted for the width of dams and adjusted the results, the differences among all categories were still highly significant (chi square = 433.54; *p* < 0.0001). However, it turned out that the activity on the dams was significantly higher than that on the trees (chi square = 302.45; *p* < 0.0001). The actual activity on the dams was also higher than expected, whereas that on the trees was lower than expected; activity on the rafts was still lower than expected (Figure 8).

## 4. Discussion

### 4.1. Method Limitations

An important part of the study was to test the suitability of kinetic sand as a medium for collecting tracks, which had not been used for that purpose before. The kinetic sand worked very well, even after several months of exposure; it did not dry out, maintained properties similar to those of wet sand, and did not lose its ability to maintain the imprinted shape, both in the case of tracks from larger animals as well as those that were very small and light. The disadvantage of kinetic sand turned out to be the initial compaction of the top layer, which disappears with prolonged pressure, i.e., the slower the animal passes, the better its tracks are imprinted. For mammals the size of a water vole and larger, this is not a major factor because they are heavy enough that even with short contact of the paws with the sand, they leave tracks. However, smaller taxa—*Apodemus* mice, *Microtus* voles, water shrews of the genus *Neomys*, and especially the smallest of Polish mammals—*Sorex* shrews and the harvest mouse *Micromys minutus* [35]—running very fast on kinetic sand may not leave any tracks at all. Due to this and the high similarity of the hind paw prints of smaller rodents to the fore and hind paw prints of shrews [31,32,33], we recorded very few tracks that are clear enough to allow identification of lower taxa, even within a family; we, therefore, created a collective category for all small mammals (micromammalia). The use of ink allows for more accurate tracks, but we stand by our opinion that under the extremely humid conditions of beaver sites, with the constant risk of flooding, kinetic sand is a more suitable tool. It did not absorb moisture from the environment, whereas plastic canopies proved to be sufficient protection against rain. If water got into the tray, the sand absorbed it, which affected its consistency and properties in these places, but only a larger amount of water was able to blur the tracks, making the sand too thin to fulfil its function.

Due to the high risk of flooding and the high activity of beavers, tunnels on dams and rafts showed the lowest performance. Beavers were particularly eager to cover tracking tunnels with wood debris and mud when they were placed closer to the centre of the dam. This might be a side effect of usual restoration works conducted by beavers, especially during periods of draughts and low water tables [36], but we cannot exclude that they target novel objects left in their territory as a threat or other kind of disturbance. This indicates that it is safer to place the tracking tunnels close to the shore, where they do not disturb beavers that much. They should also be inspected more frequently. Despite the problems mentioned above, we believe that the test of the method can be considered successful. It allows researchers to collect large amounts of data with a relatively low financial input (compared to trail cameras), it is not highly invasive for either habitat, environment, or the studied animals, and the kinetic sand can be used many times. Due to the height of the tracking tunnels (20 cm), it was, however, impossible to record mammals with a body size comparable to that of a marten (*Martes* spp.) or larger, such as otters, badgers, and canids.

### 4.2. Functions of Beaver Dams and Tree Logs for Mammals

The obtained results indicate that both beaver dams and tree logs are intensively used by small rodents and/or soricomorphs, as well as small and medium-sized mustelids. Mammal activity on dams, taking into account their width, was higher than that on trees, which partially confirms the research hypothesis. However, the species richness of mammals using tree logs was greater, mainly because of the observed differences in the use of both types of crossings. The dams were more likely to be used by rodents and shrews, probably because of the shelter they provided during crossing the watercourse. The construction of the dams allows small animals to hide between the branches forming the dam, whereas, on tree logs, they are completely exposed and vulnerable to mammalian and avian predators. Small rodents and soricomorphs may, however, use the higher dams not only as corridors for movements between both banks of a stream but also as transitional or permanent residences. This may represent a phenomenon similar to the frequent use of beaver lodges by small mammals [22], as the internal structures of both constructions are similar [36], providing comparable living conditions. Additionally, small mammals may forage among elements of the dam structure, searching for insects and other invertebrates, which are frequently eaten by both shrews and murid rodents [37,38,39]. Distinguishing among all those functions of dams was not possible in our study. It might be even harder to distinguish among them in the future, as no other available method used to record animal activity, such as trail cameras, reveals sufficient details to identify particular classes of behaviour in mammals of that body size.

Weasels, stoats, and polecats/minks (of which the latter pair have closely related tracks, making it difficult to reliably distinguish them based solely on that method, especially in habitats where both species may occur [31,32,33]) used both dams and trees. Our results show that mustelids were more willing to use trees, most likely because logs are attractive places for marking the territory by leaving faeces [40]. Elevated sites are usually selected for scent marking [41], whereas dams do not offer such opportunities as scats would become located at a lower height above the ground or water level, probably reducing the efficiency of scent dispersal. Still, mustelids do visit beaver dams regularly, presumably not only to cross water barriers but also to hunt the abundant small mammals using the same structure. Small rodents constitute the bulk of a weasel and stoat diet but are also regularly preyed upon by polecats and mink [42,43,44,45]. Some mustelid species, such as polecats, probably rely more on dams and tree logs to cross streams as they avoid entering the water when exploring banks, whereas others, such as the semiaquatic mink, enter the water and swim eagerly [43]. The only observations of the use of beaver dams by mammals published so far have not shown any watercourse crossing by small rodents with the help of these structures, whereas among mustelids, they revealed only the presence of badger, otter, and American mink, which casts doubt on the identification of species recorded by trail cameras [24]. We are not aware of any study on the use of tree logs as bridges enabling the crossing of watercourses by mammals, which makes it impossible to compare our results with those of previous works. The use of tree logs by mustelids revealed in our study seems, however, to contrast with that observed in terrestrial forest habitats. In the Białowieża Primeval Forest, during winter, weasels walk only sporadically on the upper surface of fallen trees (4% of the track length), whereas the majority of tracks are noted under the logs (67%), where they hunt rodents. Similar results were obtained for polecats and American mink, but stoats chose to walk on and under fallen trees with similar frequency [43].

Water barriers are not completely impermeable for small terrestrial rodents and soricomorphs, leaving aside semiaquatic taxa such as water voles and water shrews. All those mammals do swim effectively [46,47] and can be trapped in aquatic habitats, e.g., in pitfalls placed on a floating frame of struts [48]. In Slovakia, yellow-necked mice, *Apodemus flavicollis*, were observed to cross rivers up to 35 m in width where no natural or artificial bridges were present, whereas single bank voles, *Clethrionomys glareolus*, and field voles, *Microtus agrestis*, crossed a 3 m wide stream. However, the studied water courses did limit small mammal mobility, thus acting at least as a filter, even if not as a complete barrier [49]. We also detected the presence of small mammals at points located on the surface of open water, confirming the suitability of tracking tunnels placed on rafts as control points in our study. Astonishingly, we did not detect any mustelids on rafts, although these are considered effective methods to detect the presence of the American mink [50].

Despite the small sample size, further reduced by the loss of some tracking tunnels, the obtained data clearly support our hypothesis that beaver dams are selected by mammals, compared to the fallen trees connecting the banks of small forest rivers and streams. It seems that dams increase habitat connectivity for small terrestrial rodents, soricomorphs, and mustelids. Those constructions appear to facilitate the crossing of water barriers that usually restrict the movements of micromammalia [49]. That function may be of high significance locally, as in one case, 24 dams were found in a 1.3 km reach, whereas a density of 16 dams/km was revealed in another one [36]. On the other hand, in the case of the tiniest stream, beaver dams may only compensate for the loss of habitat connectivity caused by extensive flooding. As dams may also provide shelter and opportunity for foraging, in the final count, however, they significantly complement the improvement of habitat quality for small and medium-sized mammals in beaver-modified landscapes, already caused by the creation of wetland conditions [17,51,52]. Finally, the connective function of dams and fallen trees should not be treated as completely separate as, in many watersheds, beavers are responsible for a significant portion of logs crossing the river bed, probably equal to or higher than that caused by windbreaks or bank erosion.

## 5. Conclusions

Our study supplements the already large body of knowledge about the function of the Eurasian beaver as an ecosystem engineer. It provides a starting point for further studies on the mechanisms behind beaver impact on small mammal assemblages, an effect that has been astonishingly poorly studied. The results are also the first to quantify the use of tree logs as bridges across rivers by small and medium-sized mammals. Additionally, our study provides a useful modification of previously applied methods for collecting animal tracks in the field.

## Figures and Tables

**Figure 1 animals-13-01302-f001:**
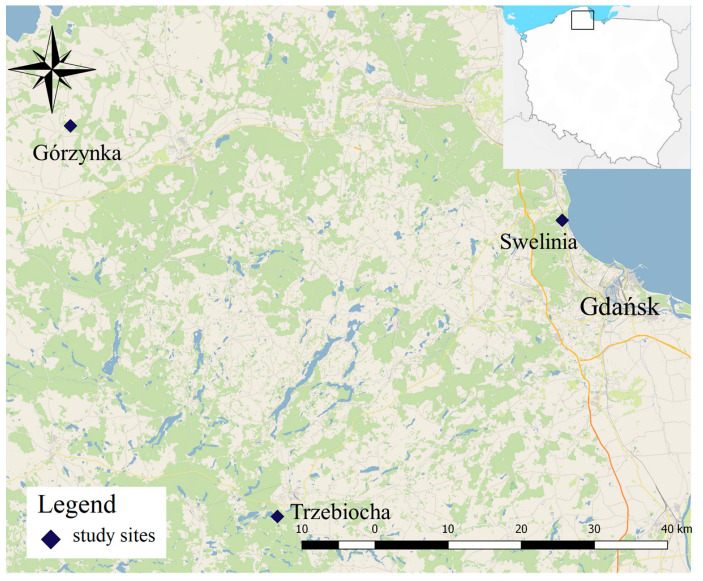
Locations of the study sites at three small streams modified by Eurasian beavers (source of background map: Open Street Map).

**Figure 2 animals-13-01302-f002:**
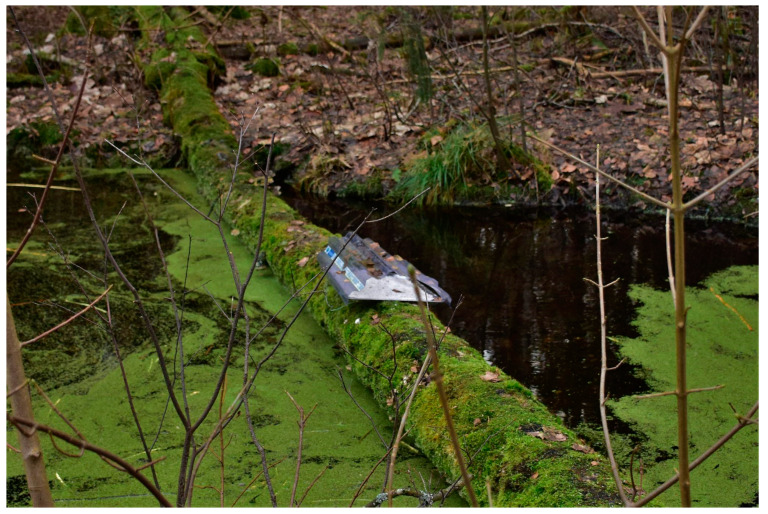
Tracking tunnel is fixed to the tree log that is connecting the banks of the stream (photo by Z. Wikar).

**Figure 3 animals-13-01302-f003:**
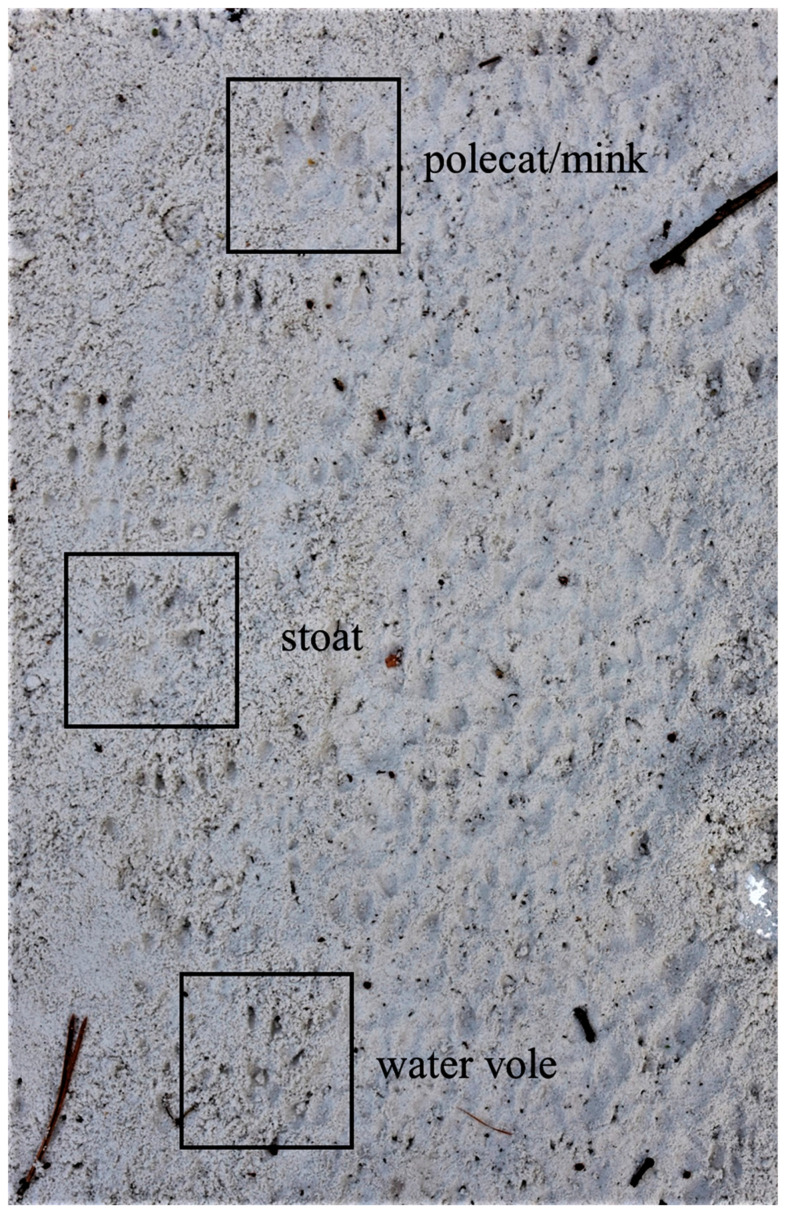
Example of a kinetic sand surface retrieved from a tracking tunnel. Three examples of single mammal tracks are indicated; 5 × 5 cm squares are given for comparison (photo by Z. Wikar).

**Figure 4 animals-13-01302-f004:**
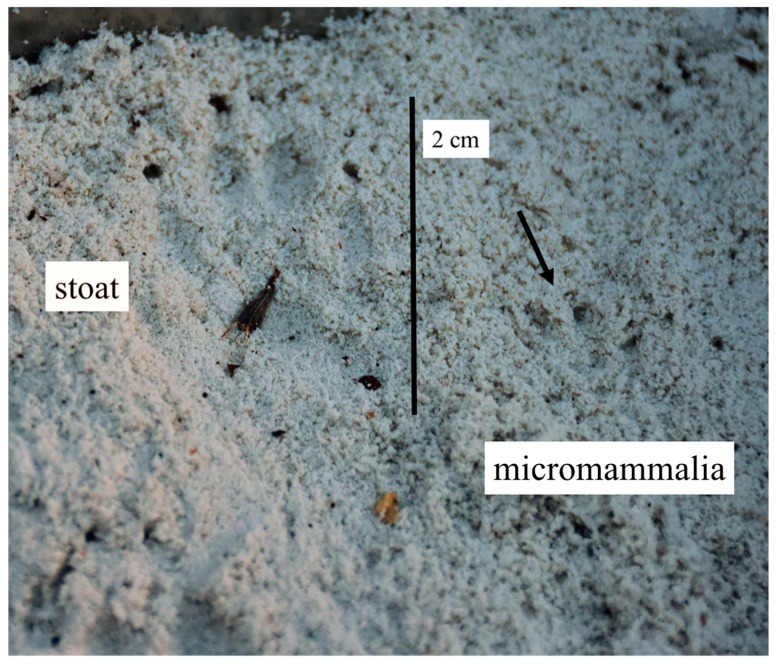
Single track of a stoat compared to that of an unidentified small rodent or shrew, classified into the collective category of micromammalia (arrow indicates claw imprints) (photo by Z. Wikar).

**Figure 5 animals-13-01302-f005:**
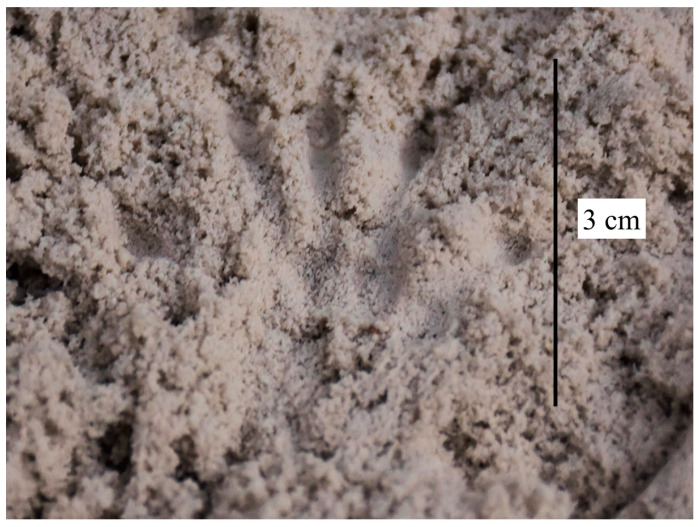
Single track of a water vole imprinted in kinetic sand (photo by Z. Wikar).

**Figure 6 animals-13-01302-f006:**
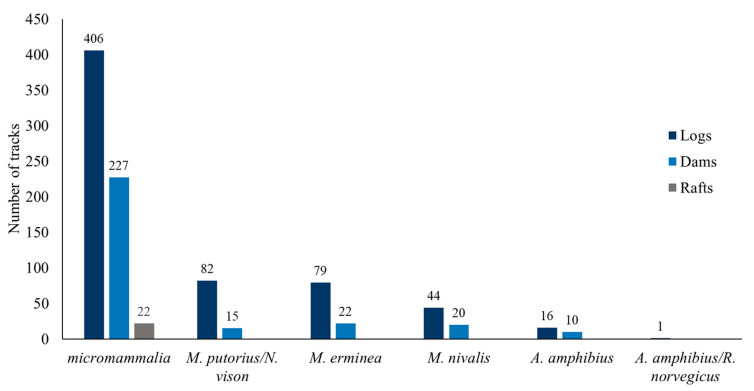
Unadjusted activity of particular mammalian taxa recorded in each category of tracking tunnel location—comparison of taxonomic composition (all sites and points combined).

**Figure 7 animals-13-01302-f007:**
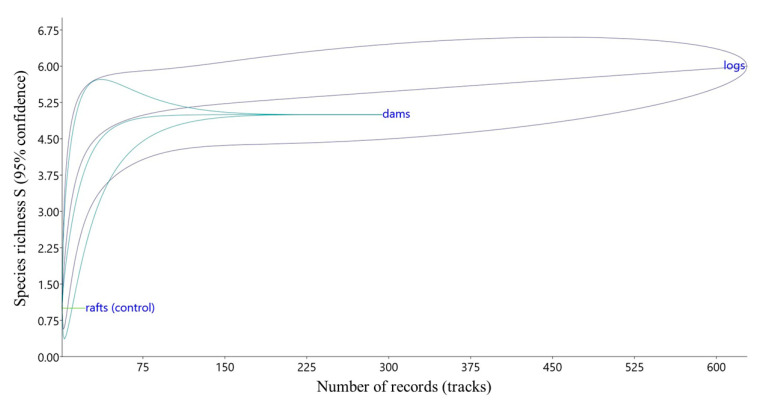
Individual rarefaction curves representing species richness (number of taxa) in particular categories of the tracking tunnels for all sites combined.

**Figure 8 animals-13-01302-f008:**
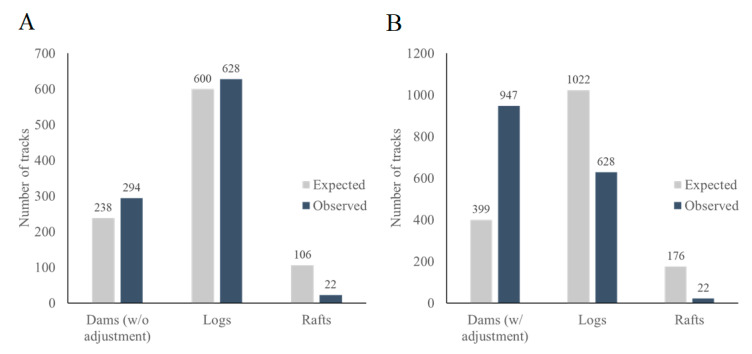
Comparison of mammalian activity among three categories of tracking tunnel locations (all taxa, points, and sites combined): (**A**) without adjustments (raw track counts); and (**B**) adjusted for the breadth of movement corridors available for animals travelling across the riverbed. The width of each dam was divided by the width of the trap (35 cm), and each obtained quotient was used as an individual coefficient by which the raw number of tracks on a particular dam was multiplied. The counts from logs and rafts remained raw.

**Table 1 animals-13-01302-t001:** Research effort and timing for the three sites. D—dam; L—log; R—raft.

Site	Number of Tracking Tunnels [D/L/R]	Date of Installation	Date of Removal	Number of Inspections
Trzebiocha	8 [1/4/3]	8 August 2020	20 December 2020	2
Swelinia	13 [3/4/6]	19 September 2020	11 December 2020	3
Górzynka	10 [4/3/3]	25 September 2020	22 December 2020	2

## Data Availability

All data are provided in the Appendix A.

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
