# Peer review of "Beaver Dams and Fallen Trees as Ecological Corridors Allowing Movements of Mammals across Water Barriers—A Case Study with the Application of Novel Substrate for Tracking Tunnels"

_animals, 2023, doi:10.3390/ani13081302_

Round 1

Reviewer 1 Report

The analysis of tracks (and signs) is a long-established, safe, valid and, above all, cost-effective method for species identification of mammals. However, the use of this method requires the ability of exact observation and determination skills. In times of DNA barcoding and camera traps, however, investigations based on tracks and signs of mammals, regrettably, have become rare. In this respect, I am very impressed about the present study!

From my point of view, there is no doubt that this work deserves publication in the journal “Animals” because it has novelty value with regard to 2 aspects: methodical and scientific. The authors describe and validate a method useful for examining small mammal movements along floating structures and point out the importance of structures that go back to the activity of beavers with regard to the spatial behavior of other mammalian species.

The study is well written, clear and concise. The methods are well described. For my taste, the figures could be improved but this is not mandatory… I have very few comments and suggestions and I congratulate the authors to this nice piece of work, which, in my opinion, adds knowledge to the field of mammal ecology.

Minor comments are:

Line 10: Please add the scientific name even in a summary.

Lines 23 and 28: Please provide scientific names also in the abstract.

Lines 37- 45: I`d appreciate the thought that there also is a tremendous variability in spatial usage within home ranges between species (inter-specific variability). Some species tend to move rather homogenously in their home range, while others tend to prefer certain structures / core areas / activity centers…Maybe added by a reference, although this might mean to renumber all following references -  I am aware of this problem caused by MDPI reference style…

Line 60: Maybe refer to lisamphibians or put “amphibians” in quotation marks…

Line 64: I was not aware of this genus name – maybe provide the respective reference: Bruce et al. 2021….

Line 69: I am not convinced whether “shrews” is the ideal translation of Soricomorpha, because, actually, shrews would be Soricidae, while Soricomorpha would be “shrew-like animals”…. Maybe you could refer to the family Soricidae here, or are there other species of Soricomorpha except from soricids occurring in your study area?

Line 90 following: I would appreciate one sentence about the advantage of tracking compared to camera traps, e.g. costs or given the likeliness that cameras could be destroyed in this experimental design…

Material & methods:

I`d really appreciate sub-headings, like study area, experimental design, analysis…

Results:

Figure 1. Please provide the source of this map. The map could greatly be improved, if you would also zoom in and provide a more detailed look on each study site.

Figure 2: Please provide the source of the photo! (applies also to figures 3-5)

Figure 3: maybe you could try and increase the contrast because it I not easy to see and compare the signs of stoat vs. mink/polecat…

Figure 6: Because of formatting, it is impossible to have a lock at the right side of this figure: my question is: is there a category “undefined”? I assume that sometimes, it was not possible to clearly identify a track?

I may have missed it: You combine the data from the three study sites and you do not put emphasis on possible differences among study sites – have you shown that there are no differences in terms of the mammal community that was identified per site? This would be a prerequisite for summarizing the data.

Discussion:

Similar to the material and methods section, I would really appreciate sub-headings. This would make the reading of the discussion more enjoyable.

Line 338: Please be careful with statement like “In the literature, there are no data at all…” Maybe there is somewhere, who knows… Better to write “to our knowledge” or “we are not aware of any study….”

Again: congrats to this nice paper!

Author Response

Thank You for all Your comments and suggestions. They really helped us to improve the manuscript.

Line 10: Please add the scientific name even in a summary.

Done

Lines 23 and 28: Please provide scientific names also in the abstract.

Done

Lines 37- 45: I`d appreciate the thought that there also is a tremendous variability in spatial usage within home ranges between species (inter-specific variability). Some species tend to move rather homogenously in their home range, while others tend to prefer certain structures / core areas / activity centers…Maybe added by a reference, although this might mean to renumber all following references -  I am aware of this problem caused by MDPI reference style…

We would not exactly agree with that statement. It is hardly to find a species homogenously using its home range, even large, marine pelagial vertebrates, especially at larger spatial scales. Even a vole living on a uniform patch of grassland, e. g. freshly mowed one, spends more time in and around its den, which results in a pattern that could be clearly interpreted as a core area. However, heterogeneity of space use is clearly dependent on heterogeneity of habitat, thus reveals interspecific variation in both heterogeneity of movement and, consequently, home range size. We adress that problem, following Your advice, providing a review of correlates of simple home range measures in ungulates, as a reference.

Line 60: Maybe refer to lisamphibians or put “amphibians” in quotation marks…

Done

Line 64: I was not aware of this genus name – maybe provide the respective reference: Bruce et al. 2021….

Done

Line 69: I am not convinced whether “shrews” is the ideal translation of Soricomorpha, because, actually, shrews would be Soricidae, while Soricomorpha would be “shrew-like animals”…. Maybe you could refer to the family Soricidae here, or are there other species of Soricomorpha except from soricids occurring in your study area?

The only ‘non-soricid soricomorph’ in our study area is European mole Talpa europaea that, as You may imagine, is hardly expected to cross the river by either dams or tree log, so the only small mammals belonging to Soricomorpha and expected to use that structures are, indeed, shrews (Soricidae). However, we would like to refer to the taxa representing the same rank, when defining term ‘micromammalia’ used in our narrative, as ‘Rodentia and Soricomorpha below a particular body size’. So, following Your advice, we cleared that out, referring to (Soricomorpha: Soricidae) as a taxonomic meaning of term ‘shrews’. I hope that would be more proper approach.

Line 90 following: I would appreciate one sentence about the advantage of tracking compared to camera traps, e.g. costs or given the likeliness that cameras could be destroyed in this experimental design…

Done, including reference

Material & methods:

I`d really appreciate sub-headings, like study area, experimental design, analysis…

Done

Results:

Figure 1. Please provide the source of this map. The map could greatly be improved, if you would also zoom in and provide a more detailed look on each study site.

Done

Figure 2: Please provide the source of the photo! (applies also to figures 3-5)

Done

Figure 3: maybe you could try and increase the contrast because it I not easy to see and compare the signs of stoat vs. mink/polecat…

Done

Figure 6: Because of formatting, it is impossible to have a lock at the right side of this figure: my question is: is there a category “undefined”? I assume that sometimes, it was not possible to clearly identify a track?

We don’t have category „unidentified” because we did not include tracks potentially belonging to birds in our study, and tracks of small mammals were included in the collective category of micromammalia anyway. We did not have trouble with identifying tracks belonging to rest of the recorded mammals.

I may have missed it: You combine the data from the three study sites and you do not put emphasis on possible differences among study sites – have you shown that there are no differences in terms of the mammal community that was identified per site? This would be a prerequisite for summarizing the data.

We expect some differences in structure of small mammal assemblage, however the data are limited. For the purpose of another research goal, we trapped small mammals in all those three sites, using pitfalls and Sherman live traps for two-three nights in every site (trap lines were located several hundred meters from the nearest tracking tunnels, pitfalls located always in riparian belt). We expected some differences, e. g. depauperated mammal assemblage in sub-urban site (indeed, none of the two water shrews was captured, and shrews in general were very scarce), presence of Neomys milleri only in Górzynka site, as it was the only site within the established range of that species (confirmed by trapping). But that differences did not affect the basic diversity indices which were very similar (species richness S => Swelinia: 6 species, Trzebiocha: 5 species, Górzynka: 6 species) and we had no data about mustelid composition, thus such expectations of differences would remain purely speculative. And, due to significant loses of tracking tunnels, our sample size diminished to a level that did not allow us to either analyse the sites separately (we add a sentence about it in Materials and Methods) or use the site as a random factor in even the crude modelling. As our study should be treated as opening of new research direction, revealing significance of dams and tree logs in maintaining habitat connectivity and providing some research protips in further projects (application of kinetic sand, method of calculating the actual use of structure by correcting the detection indices by width of the potential corridor, if necessary), we resigned from analysing the inter-site variation in the name of paper’s coherence.

Discussion:

Similar to the material and methods section, I would really appreciate sub-headings. This would make the reading of the discussion more enjoyable.

Done

Line 338: Please be careful with statement like “In the literature, there are no data at all…” Maybe there is somewhere, who knows… Better to write “to our knowledge” or “we are not aware of any study….”

Done

Once again, thank You!

Reviewer 2 Report

Nice work on the manuscript, most things are only small changes or edits. One thing to bear in mind may be that it was a little harder to follow some portions of the methods, in particular being about the ultimate fate of each of the trays/tunnels and whether the counts you provided for the number of traps were the original amount or just the number that data was collected from. One other point would be that some of your analyses, particularly those shown in figures 6-8, could have been explained a little more clearly. However, those are relatively minor and would be more for those would don't understand those data. It would be useful to have, but probably not necessary for the study itself.

Author Response

Thank You for all Your comments and advices, they allowed us to significantly improve the quality of the manuscript, thus we included the majority of them. However, the two of Your comments require some explanation from our point of view:

Line 145. Little confused how the control are floating rafts that would have to be swam to, feels like the least likely used option unless no alternatives were available.

We explained in the discussion that, based on earlier studies, it is already known that even small rodents and shrews are known to swim across small rivers, even if those structures clearly reduce their movements and small mustelids are well known to swim as well, especially semi-aquatic American mink – rafts are even considered a monitoring tool for the latter species. Thus, those rafts should not be considered useless as control points, as they reflect the original connectivity across the water barrier if no ‘solid’ corridors (like tree logs or dams) are present.

Main text of the review: One thing to bear in mind may be that it was a little harder to follow some portions of the methods, in particular being about the ultimate fate of each of the trays/tunnels and whether the counts you provided for the number of traps were the original amount or just the number that data was collected from.

We repeated the information about the original number of tracking tunnels in the Results, before the number of tracking tunnels that ‘survived’ and could be analyzed. The latter, we believed, clearly belong to the Results section only, as the test of method was a significant element of our study. We hope that repetition will allow to follow the fate of our sample of tracking tunnels, when initial and final numbers are present in one and the same sentence.

Again, thank You!